Validity and reliability of the Sustainable HEalthy Diet (SHED) index by comparison with EAT-Lancet diet, Mediterranean diet in Turkish adults

Aksoy Canyolu Burcu 1 burcu.aksoy@medeniyet.edu.tr
Martini Daniela 2
Şen Nilüfer 3
1 Department of Nutrition and Dietetics, Faculty of Health Sciences, Istanbul Medeniyet University , Istanbul , Turkey
2 Department of Food, Environmental and Nutritional Sciences (DeFENS), University of Milan , Milan , Italy
3 Department of Nutrition and Dietetics, Faculty of Health Sciences, Gazi University , Ankara , Turkey
Okpala Charles
Electronic publication date: 2024 Sep 30
Publication date: 2024
Volume: 12
Electronic Location ID: e18120
Received 2024 Mar 22; Accepted 2024 Aug 28
Copyright: © 2024 Aksoy Canyolu et al.
Copyright year: 2024
Copyright holder: Aksoy Canyolu et al.
License: This is an open access article distributed under the terms of the Creative Commons Attribution License, which permits unrestricted use, distribution, reproduction and adaptation in any medium and for any purpose provided that it is properly attributed. For attribution, the original author(s), title, publication source (PeerJ) and either DOI or URL of the article must be cited.
License URL: https://creativecommons.org/licenses/by/4.0/

Keywords: EAT-Lancet diet, Sustainable diets, Validity, Diet indexes, Diet quality

Funding: This research received no external funding.

==============================
Background

Food consumption and diet are strongly associated with sustainability. The Sustainable HEalthy Diet index was developed to measure the nutritional, environmental, and sociocultural components of sustainable diets and healthy eating patterns. However, a methodological approach has yet to be proposed for Turkish adults. This study aimed to determine the validity and reliability of the SHED index in Turkish adults.

Methods

Data were collected from 558 healthy adults using a web-based questionnaire. Internal consistency reliability was evaluated using Cronbach’s alpha coefficient, and repeatability was evaluated using the test-retest method. Construct validity was investigated using the EAT-Lancet diet and the Mediterranean Diet Adherence Screener (MEDAS), and the adapted SHED index structures’ accordance was evaluated with confirmatory factor analysis.

Results

Good reliability and repeatability were found (r = 0.758 and 0.795, respectively). A higher SHED index score was related to a greater intake of grains, fruits, and vegetables and a lower intake of meat, eggs, and dairy compared to EAT-Lancet diet food groups. A higher SHED index score was associated with a lower saturated fat and added sugar intake. While the SHED index was associated with greater adherence to the Mediterranean diet (r = 0.334, p < 0.001), it was negatively associated with non-alcoholic and diet non-alcoholic beverage consumption (r = −0.257 and −0.264, respectively; p < 0.001).

Conclusion

The SHED index showed good validity and reliability in Turkish adults. Our results suggest that the SHED index can be used in epidemiological and intervention studies because it allows the measurement of diets in terms of health and sustainability to propose adaptations accordingly.

Introduction

The incidence of diet-related chronic diseases, cancer and obesity is increasing rapidly worldwide (Rakhra et al., 2020; Alleyne et al., 2023; Gropper, 2023). At the same time, factors such as climate change, decreased freshwater reserves, reduced biodiversity, a growing world population, increased use of fossil fuels, and high greenhouse gas emissions all pose serious threats to animal husbandry, the environment, and agricultural production (Vega Mejía et al., 2018). Considering the inseparable relationship between global health, climate change, and food production (Kowalsky, Morilla Romero de la Osa & Cerrillo, 2022), the adoption of adequate, balanced, safe, and healthy diets is crucial. These diets should also be economically viable and affordable. Such measures are essential for protecting health and the environment, preventing diseases, reducing risk factors, and supporting biodiversity (United Nations Environment Program (UNEP), 2024). In this regard, recent research has tended more toward diets and their effect on health, ecosystem, and food systems. The “sustainable nutrition” approach has been identified as an effective strategy to reduce poverty and food insecurity while protecting the health of individuals and the planet (Smetana, Bornkessel & Heinz, 2019; Johnston, Fanzo & Cogill, 2014). Moreover, within the scope of the United Nations Environment Program (UNEP) (2024) in 2015, Sustainable Development Goals (SDGs) have been determined to end poverty, reduce food waste, secure safe food for all, ensure a livable world for future generations, and support biodiversity by 2030. It has been remarked that promoting sustainable diets has a critical role in achieving many of these goals (Tepper et al., 2021). The concept of sustainable nutrition was first suggested by Gussow & Clancy (1986) to promote food sustainability and eco-logical harmony. Due to the multidimensional nature of the concept of sustainability, there is no universal definition of “sustainable nutrition” yet (Sachs et al., 2023). However, it was defined by the Food and Agriculture Organization (FAO) as follows: Sustainable diets, contributing to food security for the healthy life of the current and future generations, have low environmental impacts, are protective of and sensitive to biodiversity and ecosystem, and are culturally acceptable, accessible, economically viable and affordable, nutritionally adequate, safe and healthy diets that make the best use of natural and human resources (Dernini, 2010). In a systematic review of 21 studies evaluating the potential benefits of adhering to a sustainable diet on health and environment, it was shown that a calorie-balanced diet that includes mainly plant-based food provides 60% of energy needs. It also showed that a low animal protein intake could decrease mortality and diet-related unfavourable environmental impacts (Kowalsky, Morilla Romero de la Osa & Cerrillo, 2022). Accordingly, it is clear that effective interventions are needed to promote sustainable diets, increase adherence, and raise awareness that sustainable nutrition can improve individual and community health, as well as protect resources and the environment (Johnston, Fanzo & Cogill, 2014; Tepper et al., 2021). In other words, they should be monitored quantitatively (Johnston, Fanzo & Cogill, 2014; Tepper et al., 2021). Sustainable nutrition advocates have also indicated the need for an open-source, reliable food composition database that includes data on individuals’ adherence to sustainable diets (Johnston, Fanzo & Cogill, 2014). Furthermore, measuring adherence to sustainable and healthy diets will enable politicians and authorities to understand the potential trade-offs involved in promoting these diets and to address any potential negative outcomes (Johnston, Fanzo & Cogill, 2014). Although various metrics have been proposed for this purpose, these tools do not evaluate the concepts of sustainable and healthy diets together (Béné et al., 2019; Jones et al., 2016; Żakowska-Biemans et al., 2019). Moreover, current tools are limited to only a few concepts of sustainable nutrition, such as purchasing local and organic products, sociocultural considerations, and economic factors (Jones et al., 2016; Żakowska-Biemans et al., 2019). Even though individuals’ compliance with sustainable diets is also evaluated using food frequency questionnaires (FFQ) and dietary recalls, these methods are time-consuming and require additional information, trained personnel, and equipment. They also have higher error margins (Jones et al., 2016). Therefore, there is a critical need for an instrument that can both measure and score sustainable and healthy diets while incorporating all their relevent concepts (Żakowska-Biemans et al., 2019). In recent years, several indices have been developed to measure various features of sustainable diets. These indices have been recently summarized in a scoping review of proposed sustainable diet indices (Neta et al., 2023).

Among them, Tepper et al. (2021) developed and validated the Sustainable HEalthy Diet (SHED) index in the Israeli population. With its multiple advantages, this index can be used to measure the nutritional, environmental, and sociocultural elements of sustainable diets and healthy eating patterns (Tepper et al., 2021). As a practical and evidence-based tool, the SHED index is feasible for use at the research, healthcare system, and individual levels. In addition, there was a strong correlation between SHED and the Mediterranean diet, which is acknowledged as a sustainable dietary pattern (Tepper et al., 2021). Subsequently, a study investigating the differences in the environmental footprints (land use, water use, and greenhouse gas emissions) of different diets in the Israeli population found that the SHED index was associated with the lowest greenhouse gas emissions and land use, but a higher water use (Tepper et al., 2022). This study has also shown that the index can accurately measure the nutritional and environmental components of sustainability (Tepper et al., 2022). In another study aimed at encouraging Italian university students to make healthier and more sustainable food choices, the SHED index provided a comprehensive assessment of respondents’ food-related behavior. The study considered both healthy eating and environmentally friendly attitudes, collecting information regarding the place of food purchase, use of organic food, food waste, and types of water and beverages consumed. The study found that motivation and behaviors related to healthy and sustainable nutrition enhanced the impact of the intervention (Wongprawmas et al., 2023). More recently, the SHED index has been validated in Portugal, which further supports its validity and reliability as a tool for assessing sustainable and healthy diets (Liz Martins et al., 2023).

Considering Turkey in particular, the Sustainable Development Report 2023 ranked the country 72nd among 193 countries in progress toward achieving the SDGs, indicating that the progress was inadequate (Sachs et al., 2023). The Sustainable Development Goals Assessment Report Turkey 2019, identified areas for improvement, such as improving the quality of nutrition, enhancing food diversity through domestic production, increasing agricultural productivity, reducing rural poverty, disseminating precision agriculture technologies, and managing water resources (Presidency of Strategy and Budget, 2019). A national roadmap for Sustainable Food Systems was established in 2021 to achieve the SDGs targets in Turkey (Ministry of Agriculture and Forestry, 2021). It aims to promote a sustainable food system by improving the productivity, quality, adequacy, and affordability of a healthy food supply. Ultimately, it aims to ensure food security by seeking plant and animal health and welfare requirements (Presidency of Strategy and Budget, 2019). It also promoted the role of research and innovation in achieving sustainable food systems (Presidency of Strategy and Budget, 2019). In addition, to encourage sustainable nutrition and increase awareness and adoption among the Turkish population, the Turkey Nutrition Guide 2022 now includes a sustainable diet for the first time (Ministry of health publications, 2022).

In Turkey, the sustainability of food consumption has been previously examined in terms of ecological footprint. This was done by measuring carbon footprints, greenhouse gas emissions, and water footprints, all of which are related to the use of natural resources in daily activities (Üçtug et al., 2021; Yardımcı & Demirer, 2022; Erdoğan Gövez et al., 2023). While the global ecological footprint is 2.75, Turkey’s is higher at 3.33 (Global Footprint Network, 2023). Moreover, food consumption accounts for 52% of the ecological footprint, the largest share. This highlights the need to understand the role of nutrition and dietary patterns of Turkish population in terms of sustainability (Global Footprint Network, 2012). Considering the significance of measuring and scoring healthy and sustainable diets, no methodological approach has been previously proposed for Turkish adults.

Therefore, based on these premises, the present study aimed to adapt the SHED index in the Turkish adult population by enabling the assessment of sustainable and healthy eating implementations in Turkey.

Methods

Study design and study population

This cross-sectional study was conducted between March 2023 and August 2023. It followed the tenets of the Declaration of Helsinki and was approved by Istanbul Medeniyet University Ethics Committee (Ethics Committee Decision No: E-70734980-100-2300009159). The inclusion criteria included the following: being aged ≥18 years, agreeing to participate in the study and giving written consent, not having diseases and/or disabilities that would hinder the completion of the SHED index, Mediterranean Diet Adherence Screener (MEDAS) and Water Balance Questionnaire (WBQ), not having an illness that necessitates the following of a specific diet, and not being pregnant and/or breastfeeding. It has been indicated that the sample size of at least 300 is required for a scale to be accepted as valid and reliable (Clarke & Watson, 1995). Moreover, a minimum of 30 data pairs is needed for the test-retest reliability assessment (Aksayan, Bahar & Bayık, 2002). The questionnaire was created using Google Forms and was posted on the Istanbul Medeniyet University website. The survey was advertised on social media (e.g., Facebook, LinkedIn and Instagram) and digital channels (email, WhatsApp, relevant university student groups, various research and academic groups, etc.). All adults aged ≥18 years whose mother language is Turkish were eligible to participate. Finally, 558 participants who met the inclusion criteria were included in the study. In the second stage, the SHED index was re-administered to 59 participants at a 4-week interval using the test-retest method. First, the linguistic validity of the SHED index was assessed within the scope of evaluating its psycholinguistic features. Then, its validity and reliability for the Turkish population were evaluated within the scope of its psychometric features.

Sustainable-healthy-diet index

The SHED index was developed by Tepper et al. (2021) to measure and score sustainable and healthy nutrition. It includes thirty items, each contributing with a different weight to the score (Tepper et al., 2021). Six principal elements were identified, including plant-based diet, organic awareness, drinking habits, healthy dietary consumption, consumerism and, sustainable and healthy eating unawareness (Tepper et al., 2021). The main domains of the SHED index were healthy eating (dietary intake), drinking practices (consumption of sweetened beverages and bottled water), sustainable eating (plant-based), socio-cultural aspects (organic foods), consumption of ultra-processed and plant-based foods, and environmental aspects (food waste and domestic waste practices) (Tepper et al., 2021). In detail, the main transformations include a sub-score for Healthy Eating, summarizing 10 healthy eating elements; the Sustainable Eating section, summarizing seven sustainable-eating elements; Sociocultural and socio-economic scores, which include nine purchasing patterns; ready meal scores, which includes five items; and drinking habits, also including five items. These sub-scores were defined as the summation of items from the same dimension. The 17 items of healthy and sustainability, together with the items of water score, soda, percent recycling, socio-cultural and socio-economic score, organic food consumption and ready meals and the proportion of the diet that is plant-based. This was done with the oblimin rotation, which does not force the resulting factors to be orthogonal. Before submission to the principal component analysis (PCA), all the variables were Z-transformed. Finally, the total SHED index score was calculated using PCA. Transformations were made with a minimum score of 0, maximum score of 100, negative items were reversed, and each dimension’s subscore was calculated. More details of the SHED index have been explained in the study of Tepper et al. (2021). Tepper et al. (2021), and the final SHED index was presented as Supplemental Material in that article.

Validity and reliability of the Turkish version of the SHED index

The stages performed to assess validity and reliability of the Turkish version of the SHED index are described in Fig. 1. The necessary permissions were obtained before proceeding with the assessment from Tepper et al. (2021) to use the SHED index.

Figure 1 Diagram.

Language validity

The ‘standard translation-back translation method’ recommended in the literature was used to ensure linguistic equivalence of the SHED index (Capık, Gozum & Aksayan, 2018). Firstly, a multidisciplinary group was assembled, including six researchers who were dieticians, academicians, and public health experts who knew both languages fluently and had experience in scale adaptation. The SHED index was independently translated into Turkish by three academicians and then translated back into English by three independent researchers. Subsequently, the two indexes were compared and the adapted SHED index was evaluated by a multidisciplinary team in terms of their suitability for Turkish culture and clarity. As a result of the evaluation, the ‘bottled mineral water’ and ‘bottled sparkling water’ items, which were included in the ‘drinking habits’ component of the original SHED index, were removed from the adapted SHED index. This decision was made because the consumption of such items is limited, they are not widely known by the majority of society, and they are not accessible. Moreover, by consensus of the team, some items were simplified and elaborated to ensure they are understood by individuals of all educational levels. The adapted SHED index was applied to 20 individuals who satisfied this study’s inclusion criteria but were not incorporated into the pilot study sample. The items in the index were found to be applicable and comprehensible for these individuals.

Reliability evaluation

Repeatability (test-retest method) and internal consistency reliability were evaluated to examine the psychometric features of the adapted SHED index. The intraclass correlation coefficients obtained from the test-retest in the ranges of 0.50–0.75 and 0.75–0.90 indicate moderate and good reliability, respectively, according to the literature (Koo & Li, 2016). Cronbach’s alpha coefficient should be at least 0.70 to be considered acceptable, 0.80–0.90 to be considered good, and ≥0.90 to be considered excellent (Kline, 2016).

Validity evaluation

For the purpose of examining the psychometric features of the adapted SHED index, construct validity (confirmatory factor analysis (CFA)) was evaluated and the validity of the SHED index score was assessed by comparing it with some reference diets. The CFA was used to test whether the structures of the original and adapted SHED indices were compatible. Chi-square/degree of freedom (χ2/SD), root mean square error of approximation (RMSEA), comparative fit index (CFI) and the goodness of fit index (GFI) values were reported. The χ2/SD was evaluated from <2 to <5. The CFI and GFI values of >0.9 indicate a good adaptation, and RMSEA values < 0.05 are acceptable limits (Schmitt, 2011). Moreover, while evaluating the validity of scales or questionnaires in the field of nutrition in the literature, it is recommended to determine the validity of the scale using appropriate reference methods (FFQ, diet history, isotope method, biochemical markers, and doubly labeled water) (Nelson, 2009). Accordingly, similar to the original study (Tepper et al., 2021), the SHED index score was further validated with the EAT-Lancet and the Mediterranean diets, and other dietary factors such as animal protein intake, consumption of nonalcoholic beverages, diet nonalcoholic beverages, and both tap and bottled water (derived from the Water Balance Questionnaire (WBQ)).

EAT-Lancet reference diet: The EAT-Lancet Commission (2019) constructed the first global benchmark diet with the capacity to protect health and the planet, minimize the risk of chronic disease, and maximize human well-being, drawing on all available nutritional and environmental evidence (Willett et al., 2019). This diet is rich in fruits, vegetables, and whole grains, and daily protein and fat needs are met mainly by plant-based proteins and unsaturated fats, respectively (Willett et al., 2019). The EAT-Lancet diet includes eight food groups: whole grains, tubers, starchy vegetables (potatoes and cassava), vegetables, fruits, dairy foods, protein sources (meat, eggs, fish, legumes, nuts), added fats, and added sugars (Willett et al., 2019). The recommended threshold values for each food group/food in the EAT-Lancet diet are shown in grams (Willett et al., 2019). The correlation between the total SHED index score and adherence to the EAT-Lancet reference diet was examined within the scope of validation. Adherence to the EAT-Lancet reference diet was defined as a change in the percentage of the meeting of the food groups recommendations that were met, as determined by FFQ. The relationship between the overall SHED index score and adherence to the EAT-Lancet reference diet was investigated as part of the validation process. Following the EAT-Lancet reference diet was defined as a shift in the percentage of the food categories’ recommendations that were met, as determined by FFQ.

Mediterranean Diet Adherence Screener (MEDAS): Considering that the Mediterranean diet is plant-based with moderate amounts of animal protein (Martínez-González et al., 2012), individuals with high MEDAS scores are expected to show high compliance with sustainable nutrition. Therefore, as part of the validation, the MEDAS score was compared with the total SHED index score. The MEDAS Score, which was developed by Martínez-González et al. (2012), was adapted to Turkish, and its validity and reliability were confirmed (Pehlivanoglu Ozkan, Balcioglu & Unluoglu, 2020). The instrument includes 14 questions that inquire about the following: the basic fat type used by individuals in their meals, and the amount of olive oil, fruit, vegetables, margarine/butter, red meat, sweetened beverages, wine, legumes, fish/sea products, nuts, and sugar consumed (Pehlivanoglu Ozkan, Balcioglu & Unluoglu, 2020). The score is calculated by assigning a score of 1 or 0 to each question. To clarify, if the individual consumes sufficient amounts of food in the Mediterranean diet (olive oil, fruit, vegetables, legumes, fish-sea products, nuts), one point is given if the individual does not consume 1 point and 0 points if the individual does not consume margarine-butter, red meat, sugar, or sweetened beverages or consumes them at the required level. The score range has a minimum score of 0 and a maximum score of 14. A total score of ≥ 7 is considered acceptable Mediterranean diet adherence, and a score of ≥9 is considered strict adherence to a Mediterranean diet (Koo & Li, 2016). The MEDAS score is shown in Table S2.

Water Balance Questionnaire (WBQ): The water balance questionnaire is a valid and reliable tool developed by Malisova et al. (2011) to determine the hydration status of the body. The WBQ has been adapted to Turkish and has been determined to be valid and reliable (Şen & Aktaç, 2021). Hydration status is calculated by subtracting fluid loss (urine feces and sweat) from fluid intake (food and beverages) (Malisova et al., 2011). The WBQ includes a detailed FFQ and beverage frequency questionnaire (BFQ) to determine the water intake from food and beverages. Moreover, it includes questions about the daily amount of water consumed and its type (bottled, tap water, etc.). The FFQ contains 55 foods with six frequency choices and BFQ contains 16 beverages with six frequency choices (Şen & Aktaç, 2021). A standard portion size was specified for each food in the FFQ based on the Ministry of Health Publications (2015). In this study, the BFQ in the WBQ was used to determine the amount of beverage and water types and the FFQ in the WBQ was used to determine animal protein intake. In detail, since the SHED index includes the healthy eating domain which questions compliance with a sustainable diet, such as consuming fruits and vegetables at least 400 g, avoiding meat and fatty meat products, etc. During validation, the correlation was specified between the total SHED index score and daily animal protein intake, which was derived from the FFQ in the WBQ. The adapted SHED index includes drinking habits as a main domain. The domain is divided into the following two sub-scores: the water score, which includes tap water, home water filters, and bottled water, and the soda score, which contains non-alcoholic beverages and diet non-alcoholic beverages. Within the scope of validation, the relationship was evaluated between the total SHED index score and consumption of nonalcoholic beverages, diet nonalcoholic beverages, tap water, and bottled water, all of which were derived from the BFQ in the WBQ (Malisova et al., 2011).

Statistical analysis

Data were investigated using the statistical software IBM SPSS V23 (SPSS Inc., Chicago, IL, USA). The Kolmogorov–Smirnov test was used to determine whether the data were normally distributed. Categorical variables were compared by chi-square test. Non-normally distributed data were analyzed with the Kruskal-Wallis test, and multiple comparisons were made using the Bonferroni test. Non-normally distributed and normally distributed test-retest scores were contrasted with the Wilcoxon test and Paired Sample t-test, respectively. The relationship between non-normally distributed quantitative data were evaluated with Spearman’s Rank-Order correlation coefficient and it was classified according to Dancey & John (2011). The significance level was determined as <0.05. Participants were divided into three tertiles in order to their SHED index scores. According to the total SHED index score, the lowest tertile was the 1st tertile, and the highest tertile was the 3rd tertile.

Results

Study population and characteristics of the SHED index score

A total of 558 participants completed the study. The characteristics of the sample are shown in Table 1. The sample was stratified in tertiles of the total SHED index score. Results showed no significant differences between tertiles according to education level and age. The percentage of women, vegetarians/vegans, and those following a Mediterranean diet were significantly higher in the 3rd tertile of the total SHED index score than in the 1st and 2nd tertiles.

Table 1 Characteristics of the sample in accordance with SHED index score tertiles.

	Tertiles of the SHED Indexa	Total (n = 557)	p	
1st Tertile (n = 186)	2nd tertile (n = 186)	3rd tertile (n = 185)	
Age (years)	27.4 ± 9.3	28.2 ± 9.8	28.0 ± 9.8	27.9 ± 9.6	0.532*	
Gender		
Male	46 (24.7)	27 (14.5)	14 (7.6)	87 (15.6)	<0.001**	
Female	140 (75.3)	159 (85.5)	171 (92.4)	470 (84.4)	
Education level						
Doctorate	10 (5.4)	13 (7)	15 (8.1)	38 (6.8)	0.504**	
Postgraduate	11 (5.9)	20 (10.8)	19 (10.3)	50 (9)	
Bachelor’s degree	65 (34.9)	64 (34.4)	64 (34.6)	193 (34.6)	
High school	91 (48.9)	86 (46.2)	81 (43.8)	258 (46.3)	
Secondary school	8 (4.3)	3 (1.6)	4 (2.2)	15 (2.7)	
Primary school	1 (0.5)	0 (0)	2 (1.1)	3 (0.5)	
Eating pattern		
Mediterranean diet	7 (3.8)	20 (10.8)	30 (16.2)	57 (10.2)	<0.001**	
Vegetarian/Vegan	4 (2.2)	7 (3.8)	15 (8.1)	26 (4.7)	0.019**	
Ketogenic	1 (0.5)	4 (2.2)	4 (2.2)	9 (1.6)	0.36**	
Healthy eating score	11.66 ± 3.6	15.7 ± 3.4	19.3 ± 3.4	15.5 ± 4.7	<0.001*	
Sustainable eating score	7.0 ± 3.2	10.1 ± 2.8	13.6 ± 2.7	10.2 ± 3.9	<0.001*	
Ready meals score	6.3 ± 2.5	7.1 ± 2.2	8.5 ± 1.8	7.3 ± 2.4	<0.001*	
BFV score	6.0 ± 3.8	8.4 ± 3.5	11.2 ± 4.5	8.5 ± 4.5	<0.001*	
Soda score	3.8 ± 2.3	4.8 ± 2.6	6.2 ± 1.8	4.9 ± 2.3	<0.001*	
Total SHED score	58.7 ± 50.3	56.7 ± 27.1	182.6 ± 58.6	60.0 ± 10.3	<0.001*	
Notes:

Data are expressed as means ± SD or n(%).

* Kruskall-Wallis test.

** Chi-squared test a 1st tertile ≤ 10.11 score; 2nd tertile 10.12–103.42; score; 3rd tertile > 103.43 score.

Reliability of SHED index score

Internal consistency reliability

The internal consistency reliability of the adapted SHED index (including principal components; plant-based diet, organic awareness, drinking habits, healthy dietary consumption, consumerism and sustainable and healthy eating unawareness) was specified using Cronbach’s alpha coefficient. The coefficient was 0.758 for the adapted SHED index and ranged from 0.759 to 0.814. For each of the six principal components, the coefficient values were as follows: plant-based diet (0.776), organic awareness (0.782), drinking habits (0.759), healthy dietary consumption (0.767), consumerism (0.771), and sustainable and healthy eating unawareness (0.814). Moreover, the repeatability of the index was evaluated with the intraclass correlation coefficient (Table 2). These results showed that the total SHED index score, drinking habits scores (soda, water) and plant base scores had good reliability (0.795, 0.916, 0.960 and 0.839, respectively) and sustainable eating and healthy eating scores had moderate reliability (0.589 and 0.713, respectively).

Table 2 Test-retest reliability of the adapted SHED index score and subscores.

		Mean ± SD	Median (min–max)	p	95% CI	
Healthy eating score	Test	14.51 ± 4.5	14.0 (6.0–25.0)	0.297*	0.713 [0.513–0.831] p < 0.001	
Retest	15.14 ± 5.1	15.0 (6.0–25.0)	
Sustainable eating score	Test	9.14 ± 3.8	10.0 (0.0–16.0)	0.954*	0.589 [0.303–0.758] p = 0.001	
Retest	9.11 ± 4.6	9.0 (1.0–18.0)	
Ready meals score	Test	7.19 ± 2.4	8.0 (0.0–10.0)	1.000**	1 (1-;1)	
Retest	7.19 ± 2.4	8.0 (0.0–10.0)	
Water score	Test	2.9 ± 2.8	2.0 (0.0–8.0)	0.317**	0.960 [0.933–0.976] p < 0.001	
Retest	3.0 ± 2.9	2.0 (0.0–8.0)	
Soda score	Test	5.1 ± 1.9	5.0 (0.0–8.0)	0.317**	0.916 [0.858–0.951] p < 0.001	
Retest	5.3 ± 1.9	5.0 (0.0–8.0)	
Plant based (%)	Test	8.42 ± 21.7	0.0 (0.0–80.0)	0.317**	0.839 [0.741–0.902] p < 0.001	
Retest	10.18 ± 24.9	0.0 (0.0–100.0)	
Total SHED index score	Test	38.1 ± 95.2	30.41 (−152.6 to 276.1)	0.077**	0.795 [0.652–0.879] p < 0.001	
Retest	60.0 ± 113.1	44.0 (−135.0 to 368.9)	
Notes:

* The Paired-Samples T Test.

** Wilcoxon Test, ICC 95% confidence interval: intraclass correlation coefficient.

Validity of SHED index

Construct validity

According to the result of CFA, fit indices were as follows: χ2/SD = 2.41, GFI = 0.90, CFI = 0.91 and RMSEA = 0.04, and the model fit was found to be acceptable. The confirmatory factor analysis results for the SHED index six-factor structure are presented in Table 3. Four modification processes were performed in the confirmatory factor analysis, which was created with 30 items and six factors. One item under factor six was removed from the adapted SHED index.

Table 3 Mean values and standard deviations of CFA.

Item		β 1 *	β2 **	St. Dev.	p	
Factor 1: Plant based diet	
1	<---	0.274	0.476	0.097	<0.001	
2	<---	0.462	0.802	0.109	<0.001	
3	<---	0.576	1.000			
4	<---	0.678	1.177	0.141	<0.001	
Factor 2: Organic awareness	
5	<---	0.136	0.338	0.120	0.005	
6	<---	0.403	1.000			
7	<---	0.876	2.173	0.257	<0.001	
8	<---	0.753	1.867	0.215	<0.001	
Factor 3: Drinking habits	
9	<---	0.761	1.000			
10	<---	0.225	0.296	0.065	<0.001	
11	<---	0.755	0.992	0.096	<0.001	
Factor 4: Healthy dietary consumption	
12	<---	0.504	1.000			
13	<---	0.500	0.993	0.086	<0.001	
14	<---	0.300	0.595	0.108	<0.001	
15	<---	0.668	1.327	0.144	<0.001	
16	<---	0.692	1.374	0.147	<0.001	
17	<---	0.326	0.647	0.109	<0.001	
18	<---	0.101	0.200	0.099	0.043	
Factor 5: Consumerism	
19	<---	0.249	1.000			
20	<---	0.130	0.521	0.212	0.014	
21	<---	0.226	0.908	0.250	<0.001	
22	<---	0.77	3.095	0.59	<0.001	
23	<---	0.468	1.880	0.387	<0.001	
24	<---	0.295	1.187	0.286	<0.001	
25	<---	0.550	2.212	0.440	<0.001	
26	<---	0.787	3.165	0.602	<0.001	
27	<---	0.190	0.764	0.234	0.001	
Factor 6: Sustainable and healthy eating unawareness	
28	<---	0.848	1.000			
29	<---	0.810	0.956	0.079	<0.001	
Notes:

* β1: Unstandardized beta coefficient (the factor loadings of items).

** β2: Standardized beta coefficient.

Within the scope of construct validity, the relationships between the total SHED index score and daily animal protein consumption, daily consumption of nonalcoholic beverages, diet nonalcoholic beverages, tap water and bottled water were examined. The total score was significantly negatively correlated with the consumption of non-alcoholic and diet non-alcoholic beverages (r = −0.257 and r = −0.264, respectively; p < 0.001). However, dietary animal protein intake (r = 0.066, p = 0.121) and consumption of tap water (r = −0.037, p = 0.388) and bottled water were (r = 0.036, p = 0.399) not associated with the SHED index total score (Table 4).

Table 4 The relationship between the SHED index total score with daily consumption of some beverages and animal protein.

	Total SHED score	
r	p	
Dietary animal protein intake	0.066	0.121	
Tap water	−0.037	0.388	
Bottled water	0.036	0.399	
Non-alcoholic beverages	−0.257	<0.001	
Diet non-alcoholic beverages	−0.264	<0.001	
Note:

Spearman’s Rank-Order correlation coefficient.

Associations of the SHED index with EAT-Lancet and Mediterranean diets

The construct validity of the tool was further investigated using the EAT-Lancet reference and Mediterranean diets. A higher the intake of grains, starchy vegetables, vegetables, fruits and unsaturated fats was associated with a higher SHED index-score tertile. On the other hand, a lower consumption of saturated fats and added sugars was associated with a higher SHED index-score (Table 5). Moreover, the consumption of meat, eggs, dairy and saturated fat showed high discrepancies between the score and the diet, as shown in Table 5. The consumption of meat, eggs, dairy, and saturated fat (as determined by FFQ) was much higher than the recommended intake of these foods in the EAT-Lancet reference diet for all tertiles. Grains, starchy and other vegetables, fruits, dairy, eggs and unsaturated fats were positively correlated with the score and the diet (Table 6). In Fig. 2, the mean Mediterranean diet score was 4.51 in the 1st tertile, 5.04 in the 2nd tertile, and 5.88 in the 3rd tertile. There was a moderate correlation noted between MEDAS score and the SHED index score (r = 0.334, p < 0.001). In detail, a higher Mediterranean diet score corresponded to a higher SHED index score (p < 0.001).

Table 5 Food groups of the EAT-Lancet reference diet.

	Threshold values in EAT-Lancet diet	1st tertile (n = 186)a	2nd tertile (n = 186)	3rd tertile (n = 116)		
		% EAT-Lancet	Median (min–max)	% EAT-Lancet	Median (min– max)	% EAT-Lancet	Median (min– max)	p b	
Grainsc	232	34.1	79.3 (0–475) a	39.7	92.2 (0–400) b	43.53	101.0 (1–400) b	0.005	
Starchy vegetables	50	44	22. (0–25) a	116	58.0 (0–25) b	116	58 (0–250) b	<0.001	
Vegetables	300	29	87 (0–375) a	29	87 (0–375) a	50	150 (0–375) c	<0.001	
Fruits	200	13.2	26.4 (0–300) a	34.8	69.6 (0–300) b	52.2	104.0 (0–300) c	<0.001	
Dairyd	250	97.6	244.0 (0–2,600) a	124.5	311.2 (8–2,600) b	126.4	316.0 (0–2,600) c	0.011	
Meat	14	235.7	33.0 (0–37) a	235.7	33.0 (0–375) a	235.7	33.0 (0–375) a	0.01	
Poultry	29	57.9	16.8 (0–240) a	57.9	16.8 (0–240) a	57.9	16.8 (0–240) a	<0.001	
Eggs	13	267.7	34.8 (0–150) a	267.7	34.8 (0–150) a	461.5	60.0 (0–150) b	<0.001	
Fish	28	17.7	4.9 (0–450)a	17.68	4.9 (0–150)a	17.68	4.9 (0–450)a	0.441	
Legumes	75	38.1	28.6 (0–325)a	38.1	28.6 (0–325)a	38.1	28.7 (0–325)b	0.245	
Nuts	50	13.2	6.6 (0–75) a	2.0	1.0 (0–75) b	2.0	0.9 (0–75) c	<0.001	
Unsaturated fats	40	22.7	9.1 (0–65) a	37.6	15.0 (0–62) b	38.4	15.4 (0–62) b	<0.001	
Saturated fate	11.8	277.8	32.8 (4–239) a	244.1	28.8 (5–197) b	208.5	24.6 (3–183) c	<0.001	
Added sugars	31	47.6	14.7 (0–160) a	42.7	13.3 (0–135) b	33.9	10.5 (0–160) c	<0.001	
Total energy	2.500	53.4	1.335 (1.35–8.733) a	61.8	1.545 (390–6.716) b	63.4	1.584 (233–6.648) c	0.019	
Notes:

Data are expressed in grams except for total energy expressed as kcal.

a SHED Index score: 1st tertile ≤ 10.11; 2nd tertile 10.12-103.42; 3rd tertile > 103.43.

b Kruskall-Wallis test.

c The tertile means represent the total consumption of grains, while the EAT-Lancet reference diet refers to whole grains.

d Dairy means represents whole dairy in the EAT-Lancet reference diet and kefir, yoghurt and ayran which are Turkish traditional food.

e Represents whole saturated fats except lard since it is not included in the Turkish diet.

a–c: There is no difference between tertiles with the same letter.

% EAT-Lancet: represents the adherence to the EAT-Lancet threshold amount (g) of food intake (g).

Table 6 The relationship between the SHED index total score with food groups.

Food groups of the EAT-Lancet reference diet	Total SHED score	
r	p	
Grains	0.131	0.002	
Starchy vegetables	0.319	<0.001	
Vegetables	0.339	<0.001	
Fruits	0.408	<0.001	
Dairy	0.141	0.001	
Meat	−0.095	0.025	
Poultry	−0.16	<0.001	
Eggs	0.187	<0.001	
Fish	0.038	0.372	
Legumes	0.07	0.101	
Nuts	−0.342	<0.001	
Unsaturated fats	0.201	<0.001	
Saturated fat	−0.188	<0.001	
Added sugars	−0.183	<0.001	
Total energy	0.138	0.001	
Note:

Spearman’s Rank-Order correlation coefficient.

Figure 2 Association between the SHED index score and the Mediterranean diet score.

Discussion

Undernutrition and overnutrition are significant public health problems in Turkey. Also, the prevalence and mortality of nutrition-related noncommunicable diseases are also notably high (GBD 2017 Diet Collaborators, 2019). Achieving the SDGs related to human health, diet and nutrition remains a challenge. Adopting and implementing a healthy and sustainable diet is a fundamental step toward addressing these critical problems. This can be facilitated also through quantitative monitoring of the diet to enhance health and sustainability. The SHED index has been proposed as an easy tool to measure the different pillars of sustainable healthy diets and has been already used and validated in other countries such as Israel, Italy and Portugal (Tepper et al., 2022; Wongprawmas et al., 2023; Liz Martins et al., 2023). In the present study among Turkish adults, we evaluated the validity and reliability of the SHED index. The SHED index was constructed to measure adherence to global reference diets to improve health while considering the future of planet. It also assesses adherence to the Mediterranean diet, which is recognized as a healthy and sustainable dietary pattern. Our findings indicate that SHED index was positively related to adherence to both the EAT-Lancet and Mediterranean diets, as well as plant-based food intake. Conversely, it was negatively associated with the intake of animal based foods, non-alcoholic beverages, added sugars, and saturated fat. The internal consistency reliability (Cronbach’s alpha coefficient) of the adapted SHED index was found to be acceptable for the general total shed index and all subscores as in the original study (Tepper et al., 2021). Furthermore, the repeatability (test-retest) assessment had high reliability. The total SHED index score, along with several subscores, showed higher correlations than those in the original study (Tepper et al., 2021). Specifically, the correlations for the plant based (%) score, soda score, water score, and ready meals score were 0.839, 0.916, 0.960, and 1, respectively. As a result, the adapted SHED index has high reliability and repeatability. Within the scope of construct validity, the structure of the original SHED index was first examined with CFA. All fit indices (χ2/SD, GFI, CFI, and RMSEA) were found to have appropriate values. The EAT-Lancet reference diet, MEDAS score, and WBQ (percentage of daily animal protein intake, consumption of nonalcoholic beverages, diet nonalcoholic beverages, tap water and bottled water) were used to further evaluate construct validity. The Mediterranean diet was used as it is a well renowned example of a sustainable, healthy diet that focuses not only on human health but also on environmental, economic, and social aspects (Dernini et al., 2017; Martini, 2019). In the present study, we observed a medium adherence to the Mediterranean diet (mean score 5.88), in line with previous studies performed in Turkey (Sevim et al., 2021). Although a correlation was found between the Mediterranean diet score and the SHED index score (r = 0.334, p < 0.001), similar to the original study with the Israeli population, this correlation level was lower compared to the original study (r = 0.575, p < 0.001) (Tepper et al., 2021). These results can be attributed to the fact that the Mediterranean diet adherence was assessed with different instruments in these studies. The MEDAS, for example, focuses on specific dietary factors, such as the amount of olive oil, fruit, vegetables, margarine-butter, red meat, and sweetened beverages consumed daily, etc. It does not, however, consider other components of sustainable nutrition, such as the consumption of local, organic, and traditional foods, food waste, and domestic waste streams (Trichopoulou et al., 2003). It is also important to recognize that populations have different dietary patterns; therefore, individuals’ adherence to sustainable and Mediterranean diets may differ. The correlation between the SHED index and adherence to the Mediterranean diet was higher compared to the results from the recent validation of the SHED index in Portugal (ρ = 0.406, p < 0.001) (Liz Martins et al., 2023). On the other hand, there was a negative correlation observed between the total SHED index score and consumption of non-alcoholic and diet non-alcoholic beverages (derived from WBQ), which was expected. This is consistent with the high annual consumption of these beverages in Turkey (37–40 liters) and the fact that 65% of this consumption is by individuals under 35 years of age, which aligns with the mean age of the study population (27.9 years) (Karakuş, Lorcu & Demiralay, 2016). Moreover, our results noted a lack of relationship between the SHED index score and the consumption of tap water and bottled. This lack of correlation may be due to the low consumption of tap water in Turkey, where 75% of individuals consume only bottled water, especially in urban areas (Karakuş, Lorcu & Demiralay, 2016). Regarding animal protein intake, we found a negative correlation only with poultry and eggs while no correlation was found for fish, dairy and meat. These results differ from those recently observed in Portugal where the SHED index was inversely related to the proportion of animal-sourced foods in the overall food intake (Liz Martins et al., 2023). These discrepancies can be explained by the fact that animal protein intake and ratio in Turkey is already lower than the average of the world, USA, and EU countries. Specifically, the percentage of animal protein intake relative to total daily energy intake is 39.8% in the world, 50.0% in the USA, 56.5% in the EU, and 34.2% in Turkey (Ergün & Bayram, 2021).

A correlation was found between most food groups of the EAT-Lancet reference diet and the SHED index total score, supporting the validity of the adapted SHED index to assess the adherence to sustainable healthy diets. Generally, the SHED index score was consistent with the EAT-Lancet reference diet. The highest differences between the total SHED index score and the EAT-Lancet reference diet were observed in the consumption quantities of meat, eggs, dairy products and saturated fat. This result can be attributed to the use of an FFQ for the calculation of nutrient intake, which has a high tendency to overestimate food intake. Moreover, the frequent consumption of traditional dairy products (yogurt, kefir, and ayran) that are not included in the EAT-Lancet reference diet, but are included in the Turkish diet, along with the high consumption of egg due to its affordability and accessability as a protein source may further explain these results.

Generalized sustainable diet models and generalized nutritional recommendations, such as the EAT-Lancet diet, are included in the literature. However, it is clear that a diet may be sustainable, healthy, economically just, and culturally acceptable in one country but not in another. In other words, the effects of sustainable diets on health and environmental sustainability depend largely on the country in which they are implemented. For instance, although reducing the consumption of animal-derived foods may bring sustainability benefits in high-income countries, this may not be a fair or ethical approach in low-income countries, where malnutrition is common. This aspect further underscores the importance of providing tailored adaptations of the EAT-Lancet diet while taking into consideration country-specific and culturally appropriate variations, as has been done in some previous studies (Tucci et al., 2021; Lassen, Christensen & Trolle, 2020).

On the other hand, existing research in Europe, the Middle East, and the Euro-Mediterranean as neighborhood regions of Turkey have mostly investigated the sustainability of the diet in terms of carbon footprints, water footprints, ecological footprints, footprints of dietary choices, and knowledge and attitudes of populations related to food sustainability (Laine et al., 2021; Stubbendorff et al., 2024; Hwalla, Bahn & El Labban, 2016), which is also similar in Turkey (Canyolu, Martini & Şen, 2024; Köksal et al., 2022). Most of these studies have used MEDAS to evaluate a sustainable diet and the EAT-Lancet diet has rarely been used in addition to MEDAS (Yassıbaş & Bölükbaşı, 2023; Kocaadam-Bozkurt & Bozkurt, 2023). Sustainable food consumption and chronic disease risk have also been examined. However, an index such as the SHED, which can measure both sustainable and healthy eating, has not been used before. For the first time, the SHED index has been adopted and validated in the Portuguese population in Europe (Liz Martins et al., 2023). Both Europe and Middle East and North Africa (MENA) are affected by the triple burden of malnutrition and thus food and nutrition insecurity. Therefore, adopting sustainable dietary patterns would also present an opportunity for countries to respond to the current global needs of the SDGs. Turkish validation of the SHED index can ensure the dissemination of this index and studies in the neighborhood region. The point to be emphasized here is that, to date, studies evaluating individuals’ sustainable eating patterns have not made cross-country comparisons, and the majority of studies have been conducted in high-income countries (Milner, 2018). Therefore, it is unclear which components of a sustainable diet (place of food purchase, use of organic food, food waste, and type of water and beverages consumed) should be included in country-specific recommendations. At this point, the use of tools such as the SHED index is critical. Compliance with sustainable and healthy diets in Turkish adults should first be evaluated at the national level to determine the current situation and customize the recommendations. Subsequently, based on the available data, the suitability of sustainable diets for the country and the concepts that should be customized for the country should be considered.

This study had strengths and some limitations. First, to the best of our knowledge, validation and reliability of the SHED index in the Turkish population was evaluated for the first time in this study. Second, the inclusion of a large sample across the country and the application of rigorous statistical methods, Cronbach’s alpha for internal consistency, test-retest reliability, and confirmatory factor analysis for construct validity provide credibility to our findings. Third, the SHED index was validated by comparison with the EAT-Lancet, an evidence-based global reference diet for improving human health, and the Mediterranean diet, which is accepted as a healthy and sustainable dietary pattern. Fourth, the comparison of the SHED index with other established dietary indices, the EAT-Lancet diet, and the Mediterranean diet is a strong point because it provides context and benchmarks for the SHED index. Finally, validation and reliability of the SHED index in Turkish adults provides important implications for epidemiological research and interventions in the fields of sustainability, diet, and health.

Besides these, this study has a few limitations. First, despite the large sample size, the results have limited generalizability owing to the use of a non-probability sampling approach (convenience sampling). Moreover, we included only healthy adults in this study to minimize specific factors that may affect following a sustainable diet, and excluded special groups (pregnant and/or breastfeeding women and those following a specific diet (vegans, vegetarians, and individuals who consumed only certain foods due to food intolerance, illness, or personal reasons). Second, the participants involved in this web-based study may have volunteered out of particular interest in diet, health, environment, and sustainability, potentially skewing the results concerning sustainable nutrition. In addition, due to the web-based methodology of the present study, the study sample did not cover the entire population, especially those with low literacy levels or limited Internet access. Third, the use of an online questionnaire may introduce biases, such as self-selection bias and participants’ willingness to respond accurately and honestly. In addition, not using objective measurements (such as urine and blood biomarkers) may also introduce social desirability bias. Fourth, the present study could not present socio-cultural determinants that can be related to enriching the understanding of food sustainability, since SHED index sub-dimensions (socio-cultural aspects, consumption of ultra-processed and plant-based foods, and environmental aspects) and socio-cultural components have not been evaluated.

Considering these limitations, further studies with the SHED index should be conducted with more diverse and representative sampling to provide more generalizable results and improve external validity. We suggest evaluating the validity and reliability of the SHED index in different consumer groups, especially adolescents, who are a significant segment of consumers in future studies.

We recommend conducting longitudinal studies to assess changes over time with the incorporation of objective dietary assessment methods to complement the self-reported data. We recommend that food intake be determined with more objective tools (nutrition-related biomarkers) beyond the FFQ to obtain more objective results and reduce the risk of bias. For future studies, we recommend exploring and deepening sociocultural factors using the SHED index to determine their role and impact on sustainable nutrition. We propose to compare the SHED index and its sub-dimensions with other diet indices or dietary patterns (Dietary Approaches to Stop Hypertension (DASH), Nordic and vegeterian diets, etc.), and to evaluate their performance within the scope of sustainability and sustainable nutrition.

Conclusions

Our results suggest that the SHED index is a valid and reliable instrument for comprehensively assessing a healthy and sustainable diet in Turkish adults. As a practical and evidence-based tool, SHED is feasible for use in epidemiological research and intervention studies because it allows for the measurement of diets in terms of health and sustainability. It is important to understand how the population adheres to sustainable diets to be able to propose adjustments accordingly. On the other hand, SHED can be functional in the health care system, especially for dietitians and nutritionists in their daily implications. Considering the limited data on sustainable diets in the literature, especially in low- and middle-income countries (Béné et al., 2019; Jones et al., 2016), the SHED index is valuable as it measures more than one component of sustainable diets. It can assess individuals’ food choices, eating behaviors, frequency, and preparation methods, all in terms of sustainability, in rural and urban populations in Turkey. The presence of validated tools for various countries is a crucial milestone in evaluating the sustainability of diets, as it acknowledges the unique demographics and dietary practices of each population. Thus, the evidence obtained via the SHED index can be used to assess Turkey’s progress towards achieving SDGs. Subsequently, the development of sustainable nutrition and agricultural policies by the government and non-governmental organizations can highlight how different agricultural practices affect sustainability. This can contribute to new dietary guidelines, action plans and strategies in Turkey, which may accelerate progress on key national issues such as environmental impacts of agriculture, post-harvest food supply chains, and excessive food waste. In conclusion, the SHED index is a valid tool that can promote more sustainable food consumption and systems by encouraging the achievement of international goals and supporting nutrition and health at the individual level.

Supplemental Information

Supplemental Information 1 Questions and SHED index data.

The answers given to the Mediterranean Diet Adherence Screener (MEDAS) were included. (14 questions and answers were coded as 0=No and 1=Yes).

The daily consumed amounts of the 27 foods in the Food Frequency Questionnaire are given in grams, and the daily amounts of the four beverages in the Beverage Frequency Questionnaire (BFQ) were presented in ml.

Supplemental Information 2 Responses to the SHED index.

Supplemental Information 3 SHED questions (English).

Supplemental Information 4 SHED scale (Turkish).

Authors thank Dr. Lara Naim Chehade for the English revision of the manuscript.

Additional Information and Declarations

Competing Interests

Author Contributions

Human Ethics

Data Availability

The authors declare that they have no competing interests.

Burcu Aksoy Canyolu conceived and designed the experiments, performed the experiments, analyzed the data, prepared figures and/or tables, authored or reviewed drafts of the article, and approved the final draft.

Daniela Martini analyzed the data, authored or reviewed drafts of the article, and approved the final draft.

Nilüfer Şen conceived and designed the experiments, performed the experiments, analyzed the data, prepared figures and/or tables, and approved the final draft.

The following information was supplied relating to ethical approvals (i.e., approving body and any reference numbers):

The study was conducted in accordance with the Declaration of Helsinki, and approved by the Istanbul Medeniyet University Ethics Committee (protocol code E-70734980-100-2300009159 and date of approval 14.02.2023).

The following information was supplied regarding data availability:

The raw data is available in the Supplemental Files.

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
