# Peer review of "Validity and reliability of the Sustainable HEalthy Diet (SHED) index by comparison with EAT-Lancet diet, Mediterranean diet in Turkish adults"

_PeerJ, doi:10.7717/peerj.18120_

## Round 0.1 · original submission · Minor Revisions

Kindly attend in the very best detail to the concerns raised by reviewers.

Reviewer 1 ·

Basic reporting

Dear Author,
An interesting paper makes a significant contribution to the validation of the SHED index in Turkish adults, with important implications for epidemiological research and interventions in the fields of sustainability and dietary health.
However, some limitations are worth mentioning:

Convenience sample: The study relied on a convenience sample of 558 Turkish adults, which may limit the generalizability of the results to the general Turkish population. A more representative sample could provide more generalizable results.

Online questionnaire: The use of an online questionnaire may introduce biases related to internet access and participants’ willingness to respond accurately and honestly. A combination of data collection methods could minimize these biases.

Association with other diets: Although the study compares the SHED index to the EAT-Lancet diet and the Mediterranean diet, it would be useful to include other popular diets to better contextualize the results and assess the relevance of the SHED index in various dietary contexts.

Deepening sociocultural aspects: Although the SHED index includes sociocultural components, the study does not provide in-depth details on these aspects. Further exploration of socio-cultural factors could enrich the understanding of food sustainability.
I believe that minor revisions will be necessary to make the work suitable for publication in the journal.
I believe that some revision would make this paper much stronger.
Suggestions are below.
• The author should seek professional English editing service. There are spelling mistakes.
Please add text on the limitations of your study in the Discussion section.
DISCUSSION:
This section should be enriched by results from other studies (to compare with) in eighboring countries.
Limitations about methodology should be stated.

Experimental design

The experimental design of this study has several strengths but also some areas for improvement:
Web-based surveys can introduce biases, such as self-selection bias and limitations in reaching populations with limited internet access. Combining online surveys with other data collection methods (e.g., face-to-face interviews) could mitigate these biases.
Comparative Analysis:
The comparison with established dietary indices (EAT-Lancet diet, MEDAS) is a strong point, as it provides context and benchmarks for the SHED index.
The study could enhance its comparative analysis by including additional dietary patterns or indices, offering a broader context for the SHED index's performance.
Sociocultural Factors:
The study does not provide detailed insights into how these sociocultural factors were measured and integrated. Future research should elaborate on these components to better understand their role and impact.

Validity of the findings

Strengths:

Comprehensive Approach: The study's focus on validating the SustainableHealthy Diet (SHED) Index among Turkish adults addresses an important gap in the literature. The multidimensional nature of the SHED Index, encompassing nutritional, environmental, and sociocultural components, is particularly noteworthy.

Robust Methodology: The use of a large sample size (558 healthy adults) and the application of rigorous statistical methods (Cronbach's alpha for internal consistency, test-retest for repeatability, and confirmatory factor analysis for construct validity) add credibility to the findings.

Comparative Analysis: By comparing the SHED Index with the EAT-Lancet diet and the Mediterranean Diet Adherence Screener (MEDAS), the study provides a thorough examination of its construct validity. The observed associations with dietary intake patterns and adherence to recognized healthy eating patterns strengthen the case for the SHED Index's relevance.This study lays a solid foundation for the use of the SHED Index in Turkish adults, but further research is needed to address the limitations identified. Future studies should aim to:

Use a more diverse and representative sample to improve external validity.
Provide a detailed examination of cultural adaptations of the SHED Index.
Conduct longitudinal studies to assess changes over time.
Incorporate objective dietary assessment methods to complement self-reported data.
In conclusion, while the SHED Index shows good validity and reliability among Turkish adults, addressing the highlighted limitations will be crucial for its broader application and effectiveness in promoting sustainable and healthy dietary patterns.

Reviewer 2 ·

Basic reporting

The manuscript, “Validity and reliability of Sustainable- HEalthy-Diet (SHED) Index by comparison with Eat-Lancet, Mediterranean Diet in Turkish adults,” evaluate the validity and reliability of SHED index in Turkish adults. The manuscript focuses on a very important topic and is suitable for the scope of the journal. However, there are some points that need to be corrected.

Experimental design

1. First, there are some grammatical errors in the manuscript. So, the English language/grammar of the manuscript should be reviewed. I recommend getting it fixed by a specialist in English.
2. In the method section, please mention the author's permission to study the validity and reliability of the SHED Index.
3. In the “Study design and study population” section, you refer “…. 558 randomly selected participants were included in the study…” but it should be detailed that the participants were randomly selected from where or which population.
4. In the “Sustainable-HEalthy-Diet Index” section, the index should be explained in more detail, especially in terms of the scoring.
5. The use of some words should be standardized. For example, index/scale.
6. In the “Reliability evaluation” section, you refer “… to examine the psychometric features of adapted SHED Index.” However, in the “Validity evaluation” section, you don't refer to the "psychometric features,” but the validity stage is also a part of psychometric features. Please mention the that validity stage is also a "psychometric feature.”
7. How did you examine the correlation between the SHED Index score and EAT-Lancet references, please address it in method section. Do you have an adherence score for the EAT-Lancet diet? Or did you calculate the adherence using the FFQ?
8. The section you explain the "Water Balance Questionnaire" is confusing. Please explain it more clearly.
9. The results should be written with more clarification.
10. In the “Validity of SHED index” section, you mention the relationships between the total SHED index score and daily animal protein consumption, daily consumption of nonalcoholic beverages, diet nonalcoholic beverages, tap water and bottled water. Also, there is a section as “Associations of the SHED Index with EAT-Lancet and Mediterranean diets.” Aren't these sections both for validity? Why did you mention it in separate sections?
11. A table for the “relationships between the total SHED index score and daily animal protein consumption, daily consumption of nonalcoholic beverages, diet nonalcoholic beverages, tap water and bottled water” should be added.
12. In Table 4, values, superscript letters and footnotes should be checked. For example, for starchy vegetables, there is no difference between 2nd and 3rd tertiles, but the superscript letters are different.
13. In the results, you refer the higher intake of nuts is associated with a higher SHED Index score. However, in Table 4, the 1st tertile's nuts consumption is higher than 2nd and 3rd tertiles. Which is correct?
14. In results, what do you mean by “Conversely, consumption of meat, eggs, dairy and saturated fat showed high discrepancies between the score and the diet, as shown in Table 4.” Please explain it with more clarification.
15. Lines 328-330 are duplicated in lines 330-332. Also, the sentences are the same but the references are different. Please check and confirm which is correct.
16. In Table 2, “ICC (%95 CI)” should be revised as “ICC (95% CI).”
17. In the discussion, this sentence should be revised: “In the present study among Turkish adults, we evaluated the validity and reliability of the SHED index constructed to measure adherence to global reference diet to improve health with caring the future of planet and Mediterranean diet which is acccepted as a healthy and sustainable dietary pattern.”
18. In the discussion, this sentence should be revised: “Food consumption, nutrition-related biomarkers were not measured in our study as objective outcomes of dietary intake to exclude social desirability bias and the ability of individuals to respond diet related questions about type and quantity.”
19. For Figure I, image quality should be improved.

Validity of the findings

no comment

Additional comments

no comment

·

Basic reporting

No comment

Experimental design

No comment

Validity of the findings

No comment

Additional comments

The revised manuscript is Validity and reliability of Sustainable- HEalthy-Diet (SHED) Index by comparison with Eat-Lancet, Mediterranean Diet in Turkish adults (#98201) and the aim of this study was to determine the validity and reliability of the SHED Index in Turkish adults.
Is an interesting and important manuscript. The minor observation is the following.
The information written in lines 330 to 332 us the same as the written to in lines 328 to 330.

---

## Round 0.2 · accepted · Accept

I confirm that the authors have addressed all of the reviewers' comments. The revised manuscript is acceptable for publication. Thank you authors for finding PeerJ as your journal of choice and look forward to your future scholarly contributions. Congratulations :)

Reviewer 1 ·

Basic reporting

I've checked the authors' job after the revision and I recognize they have fully addressed the raised points making the paper improved.
My final consideration is that the article is now accpetable for publication on PeerJ.
Thank you and best regards.
Kind regards.

Experimental design

The methods described with sufficient detail & information to replicate.

Validity of the findings

The conclusions are well stated, linked to original research question & limited to supporting results

Reviewer 2 ·

Basic reporting

The grammatical errors have been improved, and the statements are presented more clearly. Literature references and sufficient field background/context provided.

Experimental design

Corrections have been made, and this section has been improved.

Validity of the findings

The findings are stated clearly. Conclusions are well stated, linked to original research question.

Additional comments

The authors have made all the indicated corrections, and the manuscript is suitable for publication in the journal. They have also conducted a validity and reliability study of a useful scale for use in Turkey. I congratulate them and wish them continued success.